# Measurement of Atmospheric Muon Neutrino Disappearance Using CNN Reconstructions with IceCube †

**Shiqi Yu**  **on behalf of the IceCube Collaboration**

**Abstract:** The IceCube Neutrino Observatory is a Cherenkov detector located at the South Pole, instrumenting a cubic kilometer of ice. The DeepCore subdetector is located at the lower center of the IceCube array, and has denser configuration that has improved ability to see GeV-scale neutrinos in the detector. Convolutional neural networks (CNN) are used to reconstruct neutrino interactions in DeepCore, achieving comparable performance to the current likelihood-based method but with roughly 3000 times faster processing speeds. In this study, we present a preliminary atmospheric muon neutrino disappearance analysis using the CNN-reconstructed neutrino sample, and the sensitivity to neutrino oscillation parameter measurements is shown and compared to the recent IceCube results.

**Keywords:** neutrino oscillations; CNN; reconstruction; IceCube

## 1. Introduction

The IceCube Neutrino Observatory is a Cherenkov detector located deep under the South Pole ice. It contains 5160 digital optical modules (DOMs), constituting 78 IceCube (IC) strings and 8 DeepCore (DC) strings (Figure 1) with 60 DOMs on each string. The DC strings are in the lower center of the detector.

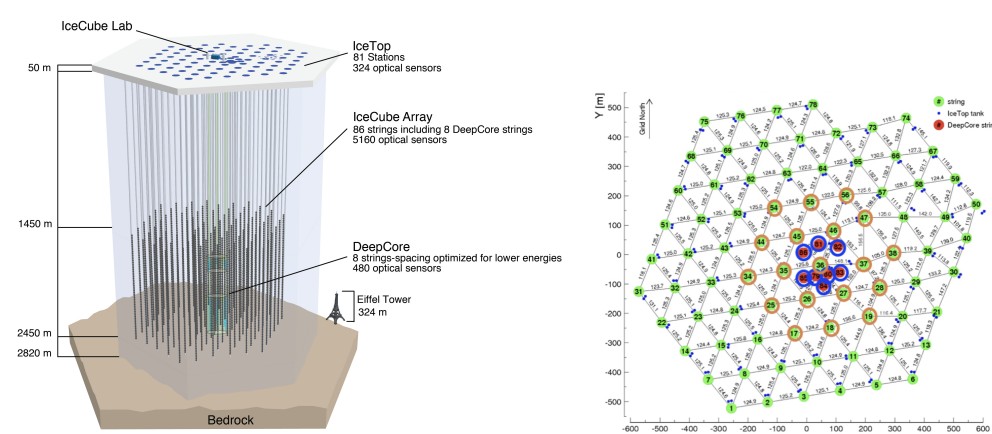

**Figure 1.** IceCube detector (**left**) and top view of detector strings (**right**) with eight DC strings (red filled).

When cosmic rays enter the atmosphere and interact with a nucleus, atmospheric neutrinos are produced. Those atmospheric neutrinos that interact in or near the IceCube detector produce charged particles. When these charged particles propagate in the ice faster than the speed of light, Cherenkov photons are emitted. The DOMs detect and convert the

light signals into digitized waveforms. Using the charge and time information extracted from the waveforms of each DOM, convolutional neural networks (CNNs) are employed to reconstruct the neutrino interactions in the detector. The DOMs in the DC strings have higher quantum efficiency and are spatially denser than DOMs in the surrounding detector. With the DC detector, neutrino interactions can be measured at the GeV scale, providing excellent sensitivity to atmospheric neutrino oscillations and in particular to atmospheric $\nu_\mu$ disappearance.

## 2. Neutrino Oscillations

Neutrinos are produced and detected in their flavor states, i.e., electron ($\nu_e$), muon ($\nu_\mu$), or tau ($\nu_\tau$), while they propagate in their mass eigenstates, i.e., $\nu_1$, $\nu_2$, and $\nu_3$. After traveling, neutrinos can be detected in different flavor states from those they were produced in; this is called neutrino oscillation, and has been studied in many experiments [1–6]. The probabilities of neutrino oscillations can be described by a unitary matrix, the Pontecorvo–Maki–Nakagawa–Sakata (PMNS) matrix [7,8], as functions of the neutrino energy ($E$) and distance traveled ($L$). The PMNS matrix can be parameterized by three mixing angles, the squared-mass difference between neutrino mass states ($\Delta m_{ij}^2$ with $i, j = \{1, 2, 3\}$), and one CP-violating phase. In $\nu_\mu$ disappearance analysis, it is possible to measure the oscillation parameters $\theta_{23}$ and $\Delta m_{32}^2$. The $\nu_\mu$ survival probability under two-flavor approximation is described by

$$P(\nu_\mu \to \nu_\mu) \approx 1 - \sin^2(2\theta_{23}) \sin^2\left(\frac{1.27\Delta m_{32}^2 L}{E}\right), \tag{1}$$

and plotted as a function of $\cos(\theta_{\text{zenith}})$ and $E$ in Figure 2, where $\cos(\theta_{\text{zenith}})$ is the neutrino arrival angle, which can be mapped into $L$. The probabilities of neutrino oscillations are functions of ($L/E$), which suggests that precise reconstruction is critical in neutrino oscillation measurements. CNNs are employed to reconstruct neutrino interactions in and near the DC detector using the digitized pulses of neutrino events.

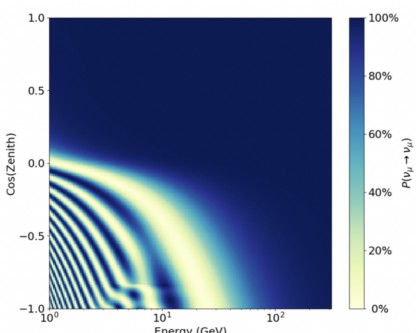

**Figure 2.** The $\nu_\mu$ disappearance probability as a function of the energy and $\cos(\theta_{\text{zenith}})$.

## 3. Convolutional Neural Networks

CNNs have been broadly used in neutrino experiments for regression and classification purposes [9,10]. All of the DC strings along with the surrounding nineteen IC strings are used to learn the features of the neutrino interactions in and near the DC detector. Five summarized variables (the total charge, times of the first and last charges, time-weighted mean of the charge, and time-weighted standard deviation of the charge) are calculated from the digitized waveforms. Because the IC and DC strings are configured differently, they are fed to the CNN through two separate input layers, allowing the local features to be better learned and used by the CNN.

## 4. Reconstruction

### 4.1. Training

To build the $\nu_\mu$ disappearance analysis, it is necessary to reconstruct the neutrino interactions in the DC detector and select those neutrino events in the selected sample that are well reconstructed with a reasonably high signal-to-background ratio. We employ the CNN to reconstruct the neutrino energy, arrival angle ($\theta_{zenith}$), interaction vertex, particle identification (PID), and a classifier for cosmic ray muon rejection. The energy, $\theta_{zenith}$, and PID are used together as the analysis observables, while the event interaction vertex and muon classifier are respectively used to select the events in or near the DC detector and to further reduce the background muon rate. The PID can be inferred from the outgoing particle trace of the event in the detector. Most of the signal ($\nu_\mu$ CC) events leave a long muon track in the detector, and as such are track-like. The other neutrino interactions, such as $\nu_e$ CC, NC, and most $\nu_\tau$ CC, produce relatively short particle cascades, and are considered 'cascade-like'. For reconstruction purposes, different training samples are used to optimize the reconstruction performance.

As shown in Figure 3, for the energy, $\theta_{zenith}$, and interaction vertex we use Monte Carlo (MC) simulated $\nu_\mu$ CC events with a relatively flat distribution of the corresponding variable to train the CNN. In this way, the reconstruction performance is optimized on the signal events for the best reconstruction resolution while ensuring that the CNN reconstructions have reasonable performance on the background events ($\nu_e$ CC, $\nu_\tau$ CC, and NC events). The neutrino energy and interaction vertex is trained on the flat true energy $\nu_\mu$ CC sample, which has 7 million events in total, split into 80% for training and 20% for testing. The $\theta_{zenith}$ is trained on a $\nu_\mu$ CC sample flattened in true $\theta_{zenith}$ distribution. The PID is trained on a balanced mixture (50:50) of track ($\nu_\mu$ CC) and cascade ($\nu_e$ CC and $\nu_\mu$ neutral-current) events with a total of 6 million events. The muon classifier is trained on approximately 4 million events with a combination of $\nu_e$ CC (20%), $\nu_\mu$ CC (40%), and atmospheric muons (40%).

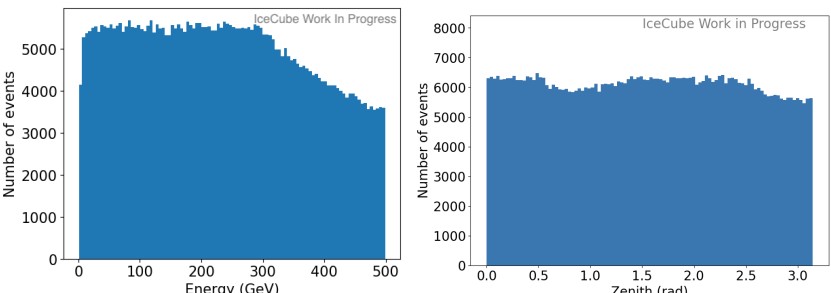

**Figure 3.** Training samples for energy (**left**) and $\theta_{zenith}$ (**right**).

### 4.2. Testing

Performance is evaluated using the nominal MC $\nu_\mu$ CC and $\nu_e$ CC events with flux, cross-section, and oscillation weights applied. The baseline benchmark is drawn using the current likelihood-based reconstruction [11].

As shown in Figure 4, the fractional error of energy reconstructed by the CNN has a smaller bias along the true neutrino energy compared to the likelihood-based method, especially in the low energy region; both methods show comparable performances on the reconstructed $\theta_{zenith}$, PID, and muon classifier. For the latter two, the area under curve (AUC) is used as the performance metric, where a larger AUC means a better performance. The reason for CNN's better energy resolution in the low-energy region is that the focus is on the near DC region and it is optimized for low-energy events, which contribute the most to the oscillation sensitivities.

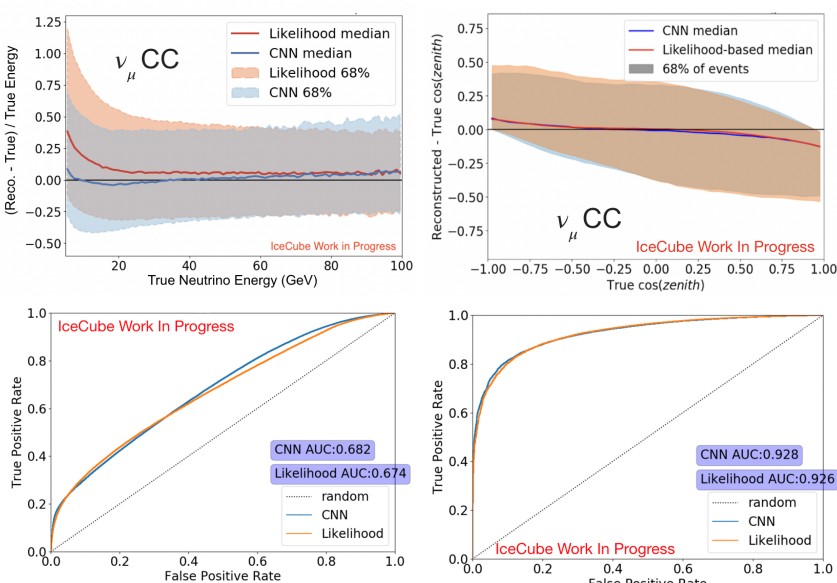

**Figure 4.** Performance comparison of CNN (blue) and likelihood-based (orange) on reconstruction of (from top left to bottom right) the energy, $\theta_{\text{zenith}}$, PID, and muon classifier, where random represents the baseline performance of particles being randomly assigned to different categories.

Figure 5 shows the purity of the selected samples by using the interaction vertex to select events starting inside or near the DC detector. The fraction of events selected by the CNN-reconstructed vertex that truly start in the DC region is similar to that of the likelihood-based method. For events that are far from the DC detector, the CNN has worse performance than the likelihood-based method, which is largely due to the information loss in the inputs to the two methods. Farther from the near-DC region, the likelihood-based method makes use of all photon information from the entire detector, meaning that it has more accurate estimates of the vertex position. However, because we have sufficiently large statistics in our data, the loss in uncontained events is unimportant to this analysis.

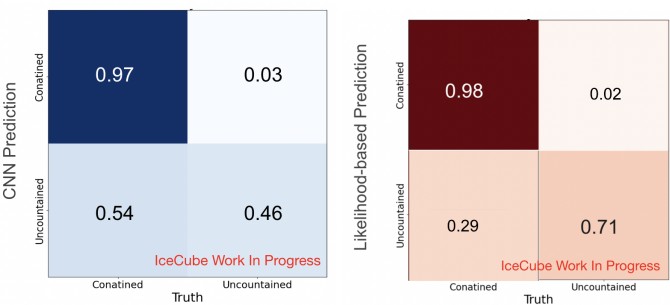

**Figure 5.** Selected sample purity when using CNN (**left**) and likelihood (**right**) methods reconstructed interaction vertex; the rows sum to 1, while the numbers corresponds to the fraction of unweighted events.

### 4.3. Processing Speed

With comparable or slightly better performance, CNN reconstruction runs approximately 3000 times faster than the likelihood-base method (Table 1) assuming the use of 1000 CPUs, and is even faster when running on GPUs. Faster processing speeds represent a major advantage for atmospheric neutrino datasets, where full MC simulation is approximately at the order of $10^8$ events.

**Table 1.** Averaged processing speed of the CNN and likelihood-based methods assuming 1000 cores.

|  | Time per 3k Events (s) | Full Sample (1000 Cores) |
|---|---|---|
| CNN on GPU | 21 | 13 min |
| CNN on CPU | 45 | 7.5 h |
| Likelihood-based on CPU | 120,000 | 46 days |

## 5. $\nu_\mu$ Disappearance Analysis

From the CNN reconstruction, we can construct the $\nu_\mu$ disappearance analysis to measure $\theta_{23}$ and $\Delta m_{32}^2$. The energy, $\cos(\theta_{zenith})$, and PID are used for analysis binning, as shown in Figure 6, providing better sensitivity to physics parameter measurements and robustness against systematic uncertainties. As shown in Figure 7, events with PIDs in the range of [0.55, 1] are placed in the track-like bin, events with PIDs in the range of [0, 0.25] are placed in the cascade-like bin, and the middle histogram (the mixed bin) contains the remaining events.



**Figure 6.** Preliminary sample in cascade (**left**), mixed (**middle**), and track (**right**) bins with flux, cross-section, and oscillation weights applied; the pink circle highlights the $\nu_\mu$ disappearance "valley".

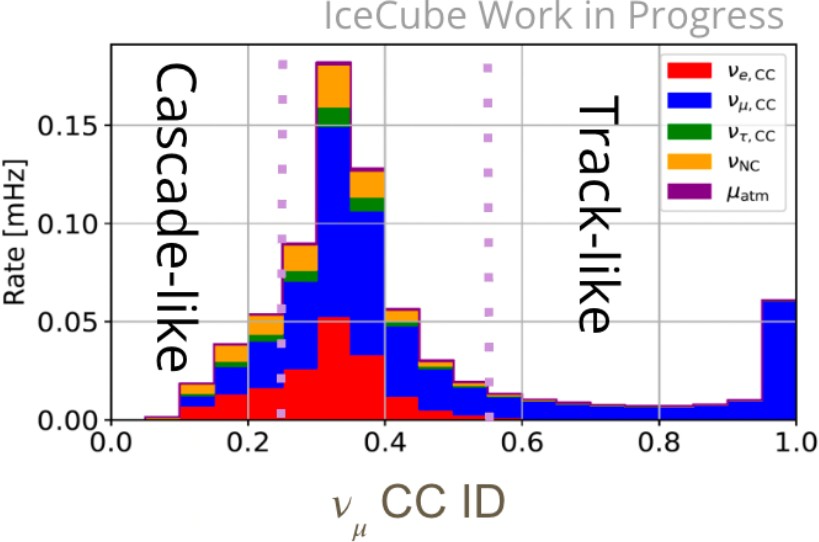

**Figure 7.** MC distribution of CNN-reconstructed PIDs with stacked interaction types (colors) and boundaries used for binning (dashed).

As shown in Figure 8, the projected sensitivity contour using CNN reconstruction shows significant improvement in comparison to the DeepCore 2021 result, and is at a level similar to using likelihood-based reconstruction. The projected sensitivity contour is compatible with previous experimental results [3–5,12].

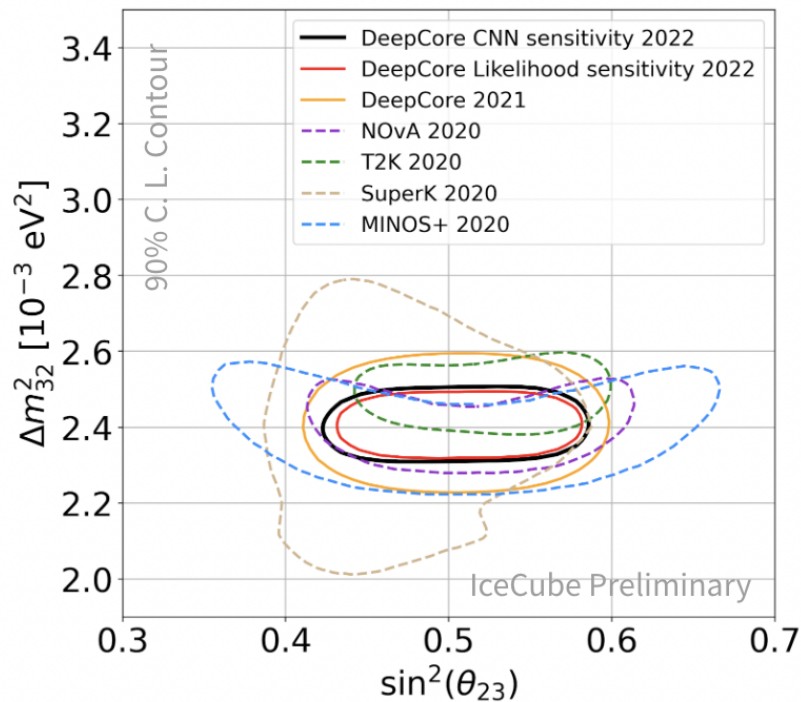

**Figure 8.** Preliminary sensitivity using CNN reconstruction (black) projected from DeepCore 2021 result (yellow) compared to using likelihood-based reconstruction (red).

## 6. Summary

In this analysis, we achieve comparable reconstruction performance to the likelihood-based method when employing CNNs, except with processing speeds up to 3000 times faster. When using the preliminary sample selected and binned by the CNN-reconstructed variables, the sensitivity to $\theta_{23}$ and $\Delta m_{32}^2$ measurement is been improved relative to the DeepCore 2021 results, and is comparable to using likelihood-based reconstruction. The CNN method can be adapted and applied to the future detector (the IceCube Upgrade [13]) to further improve the measurement precision of neutrino oscillation parameters.

**Funding:** The authors gratefully acknowledge the support from the following agencies and institutions: USA—U.S. National Science Foundation-Office of Polar Programs, U.S. National Science Foundation-Physics Division, U.S. National Science Foundation-EPSCoR, U.S. National Science Foundation AI Institute for Artificial Intelligence and Fundamental Interactions, U.S. National Science Foundation-Office of Advanced Cyberinfrastructure, Wisconsin Alumni Research Foundation, Center for High Throughput Computing (CHTC) at the University of Wisconsin–Madison, Open Science Grid (OSG), Partnership to Advance Throughput Computing (PATh), Advanced Cyberinfrastructure Coordination Ecosystem: Services & Support (ACCESS), Frontera computing project at the Texas Advanced Computing Center, U.S. Department of Energy-National Energy Research Scientific Computing Center, Particle astrophysics research computing center at the University of Maryland, Institute for Cyber-Enabled Research at Michigan State University, Astroparticle physics computational facility at Marquette University, NVIDIA Corporation, and Google Cloud Platform; Belgium—Funds for Scientific Research (FRS-FNRS and FWO), FWO Odysseus and Big Science programmes, and Belgian Federal Science Policy Office (Belspo); Germany—Bundesministerium für Bildung und Forschung (BMBF), Deutsche Forschungsgemeinschaft (DFG), Helmholtz Alliance for Astroparticle Physics (HAP), Initiative and Networking Fund of the Helmholtz Association, Deutsches Elektronen Synchrotron (DESY), and High Performance Computing cluster of the RWTH Aachen; Sweden—Swedish Research Council, Swedish Polar Research Secretariat, Swedish National Infrastructure for Computing (SNIC), and Knut and Alice Wallenberg Foundation; European Union—EGI Advanced Computing for research; Australia—Australian Research Council; Canada—Natural Sciences and Engineering Research Council of Canada, Calcul Québec, Compute Ontario, Canada Foundation for Innovation, WestGrid, and Digital Research Alliance of Canada; Denmark—Villum Fonden, Carlsberg

Foundation, and European Commission; New Zealand—Marsden Fund; Japan—Japan Society for Promotion of Science (JSPS) and Institute for Global Prominent Research (IGPR) of Chiba University; Korea—National Research Foundation of Korea (NRF); Switzerland—Swiss National Science Foundation (SNSF).

**Institutional Review Board Statement:** Not applicable.

**Informed Consent Statement:** Not applicable.

**Data Availability Statement:** Not applicable.

**Conflicts of Interest:** The author declares no conflict of interest.

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
