# Peer review of "Measurement of Atmospheric Muon Neutrino Disappearance Using CNN Reconstructions with IceCube†"

_psf, doi:10.3390/psf8010062_

Round 1

Reviewer 1 Report

The introduction is a bit sloppy. For example the relationship between flavor and mass states should be discussed and the PMNS formalism introduced. This would allow the author to introduce the ideas of a 3D rotation necessitating the 3 rotation angles which are never properly introduced. It is also best to explain that the oscillations are a function of time of flight which is approximated by Dm^2*(L/E). The facts the Eqn (1) is a 2-flavor approximation without matter effects included should also be mentioned.  

In Sec 4.1, can you please explain why NC and CCnutau topologies are not used in the training samples?

Figure 6: There must be a better way to present this information. The given plot is hard to interpret and I have no idea what the impact is to the bottom line analysis sample changes between the two classifiers. Something that provide information on the backgrounds, etc, like a table of the number of events in each signal and background category in the selected samples. It looks like the "uncontained" classification gets significantly worse, but this is not mentioned. If it not important than do not show it, and it it is important, discuss.

Why are there no plots of the CNN outputs, fora example the flavor discriminant?

It would also be interesting to hear about events that had a significant shift in their reco properties or change in PID from the likelihood method to the CNN. Where there certain event features that correlated with large changes in characterization? I can see, for example, that the reco energy bias changes significantly for lower energy neutrinos. Why was this population affected, etc?

Figure 7: While these 3 plots are nice they are hard to interpret (and to understand the source of osc param sensitivity) without seeing separately the flux, xsec, efficiency/acceptance, and oscillogram in this binning. Seeing the PID spectrum before this wold also make the PID range cuts given in the text more useful in interpreting the results.

Some discussion of how uncertainty propagation would work with the CNN vs the likelihood method would be interesting. 

I know this is a proceeding, so there is likely some strict page limits. Therefore I understand if you cannot include all the changes I am asking about. However, I do not think Figs 3 and 4 add much to the paper. They can likely be combined or removed to open up more room for results plots.
